# Gossypitrin, A Naturally Occurring Flavonoid, Attenuates Iron-Induced Neuronal and Mitochondrial Damage

**DOI:** 10.3390/molecules26113364

**Published:** 2021-06-02

**Authors:** María Ángeles Bécquer-Viart, Adonis Armentero-López, Daniel Alvarez-Almiñaque, Roberto Fernández-Acosta, Yasser Matos-Peralta, Richard F. D’Vries, Javier Marín-Prida, Gilberto L. Pardo-Andreu

**Affiliations:** 1Center for Research and Biological Evaluations, Institute of Pharmaceutical and Food Sciences, University of Havana, 222 St. #2317 b/ 23 and 31, La Coronela, La Lisa, La Habana CP 13600, Cuba; mabecquer@ifal.uh.cu (M.Á.B.-V.); adonis.armenteros@ifal.uh.cu (A.A.-L.); dani@ifal.uh.cu (D.A.-A.); javier.marin@ifal.uh.cu (J.M.-P.); 2Pharmacy Department, Institute of Pharmaceutical and Food Sciences, University of Havana, 222 St. #2317 b/23 and 31, La Coronela, La Lisa, La Habana CP 13600, Cuba; roberto.fernandezac91@gmail.com; 3Bioinorganic Laboratory, Faculty of Chemistry, University of Havana, Zapata and G, Vedado, La Habana CP 10400, Cuba; yasser.matos@fq.uh.cu; 4Facultad de Ciencias Básicas, Universidad Santiago de Cali, Calle 5 # 62-00, Cali CP 76001, Colombia; richard.dvries00@usc.edu.co

**Keywords:** gossypitrin, iron, HT-22 cells, mitochondria, neurodegeneration, neuroprotection

## Abstract

The disruption of iron homeostasis is an important factor in the loss of mitochondrial function in neural cells, leading to neurodegeneration. Here, we assessed the protective action of gossypitrin (Gos), a naturally occurring flavonoid, on iron-induced neuronal cell damage using mouse hippocampal HT-22 cells and mitochondria isolated from rat brains. Gos was able to rescue HT22 cells from the damage induced by 100 µM Fe(II)-citrate (EC_50_ 8.6 µM). This protection was linked to the prevention of both iron-induced mitochondrial membrane potential dissipation and ATP depletion. In isolated mitochondria, Gos (50 µM) elicited an almost complete protection against iron-induced mitochondrial swelling, the loss of mitochondrial transmembrane potential and ATP depletion. Gos also prevented Fe(II)-citrate-induced mitochondrial lipid peroxidation with an IC_50_ value (12.45 µM) that was about nine time lower than that for the *tert*-butylhydroperoxide-induced oxidation. Furthermore, the flavonoid was effective in inhibiting the degradation of both 15 and 1.5 mM 2-deoxyribose. It also decreased Fe(II) concentration with time, while increasing O_2_ consumption rate, and impairing the reduction of Fe(III) by ascorbate. Gos–Fe(II) complexes were detected by UV-VIS and IR spectroscopies, with an apparent Gos-iron stoichiometry of 2:1. Results suggest that Gos does not generally act as a classical antioxidant, but it directly affects iron, by maintaining it in its ferric form after stimulating Fe(II) oxidation. Metal ions would therefore be unable to participate in a Fenton-type reaction and the lipid peroxidation propagation phase. Hence, Gos could be used to treat neuronal diseases associated with iron-induced oxidative stress and mitochondrial damage.

## 1. Introduction

Iron is an essential cofactor in many crucial biological processes such as DNA synthesis and restoration, oxygen transport, cellular respiration, xenobiotic metabolism, and hormone synthesis [1]. However, at high intracellular levels, it can increase cellular reactive oxygen species with a modification in the cellular redox potentials [2,3]. Iron dyshomeostasis is observed in almost every neurological condition such as Parkinson’s disease (PD), Alzheimer’s disease (AD), Huntington’s disease (HD), multiple sclerosis, Friedreich’s ataxia, epilepsy, restless legs syndrome, and stroke [4,5]. Particularly, in intracerebral hemorrhage, where there is a rupture of the microvasculatures and the blood released produces a blood-derived iron exposure of neural cells, the main mechanisms of neurodegeneration are iron- and heme-induced ROS production, inflammation, and excitotoxicity [6,7]. In vitro and in vivo experimental models of brain hemorrhage have shown the presence of ferroptosis, an iron-dependent form of regulated necrosis [8]. Furthermore, increasing evidence suggests that the disruption of iron homeostasis is essential in the loss of mitochondrial function in neural cells [9,10], which could in turn lead to neurodegeneration [11,12,13,14]. A relationship was recently reported between the early-onset form of Parkinson’s disease with iron metabolism dysregulation, and mitochondrial impairment [15]. The co-existence of relatively high amounts of both iron and ROS in the mitochondrial compartments makes this organelle susceptible to hydroxyl radical-mediated damage. Pharmacological agents targeting brain iron regulation and iron-mediated mitochondrial impairment may therefore be therapeutically useful against neurodegeneration. Several iron-chelating agents have been found to exhibit neuroprotective actions in different experimental models [16,17,18,19]. On the other hand, there have been many reports on the neuroprotective potential of naturally occurring polyphenols due to their antioxidant actions based on a combination of iron chelation and free radical scavenging activities [20,21]. Our group has observed that catechol-containing polyphenols like mangiferin, guttiferone-A, and rapanone interact with iron, both in vitro and in vivo, thus annulling its catalytic role in promoting ROS [22,23,24,25,26,27,28]. We also recently reported the potent antioxidant effects of gossypitrin (Gos), a naturally occurring flavonoid (Figure 1), indicating its potential use as a neuroprotective agent [29]. Nevertheless, little is known of the mechanisms involved in these antioxidant effects. The presence of a planar six-member cyclic system with electron delocalization, a catechol moiety, and three more aromatic hydroxyl groups (two of them adjacent to a carbonyl group), strongly suggest that this molecule could act as an iron-chelating agent, since these are structural features that favor iron chelation by flavonoids [30]. Hence, the purpose of this study is to evaluate the ability of Gos to protect HT-22 neuronal cells and isolated rat brain mitochondria from iron-mediated oxidative damage. Other studies in cell/mitochondria-free systems were carried out to further characterize the nature of the Gos-iron interaction.

## 2. Results

### 2.1. Gos Rescues HT-22 Cells from Fe(II)-Induced Cell Death

Fe(II) (100 µM) reduced HT-22 cell survival to 17.4% after 24 h of exposure, as estimated by the MTT assay. Gos prevented hippocampal cell death in a dose-dependent manner. The EC_50_ was 8.64 ± 1.58 µM (Figure 2A, inset). Full cell protection was observed with 100 µM Gos that was more effective than Trolox (500 µM), the well-known hydrosoluble antioxidant, which prevented cell death by 74%.

HT-22 exposure to iron for 2 h (prior to cell death execution) induced the dissipation of the mitochondrial membrane potential, as estimated by the retention of rhodamine 123 within the organelles (Figure 2B), and ATP depletion (Figure 2C). This suggests a possible association between the toxic iron-mediated effects on the mitochondria and cell viability. Gos prevented the iron-induced dose-dependent impairment of the mitochondria, following the patterns of neuron viability which showed that it may prevent iron-induced HT-22 cell death through mitochondrial protection.

### 2.2. Gos Rescues Rat-Brain Mitochondria from Iron-Induced Mitochondrial Lipoperoxidation

Brain mitochondria incubated with Fe(II)-citrate (50 µM) underwent a marked lipoperoxidation expressed by an increase in the formation of the thiobarbituric acid reactive substance (Figure 3A). Gos inhibited the iron-mediated mitochondrial lipid peroxidation in a concentration-dependent manner (IC_50_ value of 12.45 ± 1.24 µM) (Figure 3A, inset). The effects of Gos on *tert*-butyl hydroperoxide (300 µM)-induced mitochondrial lipid peroxidation (IC_50_ value >100 µM) shows that it is more effective in preventing iron-induced lipid peroxidation than peroxyl induced lipid peroxidation, suggesting its ability to act directly on iron (Figure 3B).

### 2.3. Gos Prevents Iron-Mediated Mitochondrial Swelling, ΔΨ Dissipation, and ATP Depletion

Figure 4A shows that 50 µM Fe(II)-citrate complex induced mitochondrial swelling as expressed by a decrease in the absorbance at 540 nm (line f versus line a). It was associated with a spontaneous and complete ΔΨ depolarization (Figure 4B, line f versus line a), and ATP depletion (Figure 4C, Fe(II)/citrate column). Gos inhibited the iron-induced swelling process and ΔΨ dissipation in a concentration-dependent manner (Figure 4A,B lines b–e). The absorbance values (mean ± SD at 7 min) from Panel A were: 0.482 ± 0.013 (line a), 0.459 ± 0.011 (line b), 0.445 ± 0.007 (line c), 0.435 ± 0.015 (line d), 0.360 ± 0.011 (line e), and 0.338 ± 0.012 (line f). The fluorescence values from Panel B (mean ± SD at 7 min) were 15.03 ± 1.71 (line a), 19.41 ± 2.07 (line b), 22.29 ± 2.58 (line c), 45.505 ± 4.47 (line d), 118.468 ± 5.76 (line e), and 146.201 ± 5.83 (line f). Statistically significant differences were found between line f and the other lines, at *p* < 0.05. The naturally occurring flavonoids also prevented iron-mediated ATP depletion in a concentration-dependent manner (Figure 4C). A strong interaction of Gos with iron is suggested as the key mechanism for neuroprotection against iron-induced damage, both in HT-22 neuronal cells and in brain mitochondria. To further characterize this Gos-iron interaction, different experiments on a cell/mitochondria-free system were performed.

### 2.4. Gos Induces Fe(II)-Citrate Autoxidation and O_2_ Consumption

Gos decreased Fe(II) concentration (1-min preincubation) in a reaction medium containing 2 mM citrate (without mitochondria) in a dose-dependent manner (Figure 5A, black circles). A 5-min preincubation period (Figure 5A, red boxes) decreased Fe(II) concentration even more. Gos also increased the rate of O_2_ consumption, possibly due to oxidation of Fe(II) to Fe(III) (Figure 5B,C). These results suggest that Gos could be removing Fe(II) from the citrate complex and oxidizing it to the ferric form in a process that requires O_2_ as the electron acceptor. Accordingly, Gos could be hindering the formation of ^●^OH radicals by inactivating the Fe(II) necessary for the Fenton-Haber-Weiss-type reaction, which can initiate the lipid peroxidation of the mitochondrial membrane.

### 2.5. Gos–Fe(II) Complexes Detection

A plausible hypothesis to explain the above results is that Gos forms a transient complex with Fe(II) that facilitates its oxidation by oxygen. This transitional complex could deliver its electrons more readily than Fe(II)–citrate, generating a more stable complex with Fe(III). Figure 6A shows a typical spectrum of Gos with maximum absorption at 276, 332 and 380 nm (black line). The addition of Fe(II) induced a concentration-dependent decline in the maximum absorption peaks and the appearance of a new one near 500 nm (Figure 6A). The occurrence of an array of spectra originated from the Gos spectrum was confirmed by the presence of an isosbestic point at λ_iso_ = 405 nm (see black dots), indicating a chemical equilibrium (complexation) between free and complexed Gos. The inset (Figure 6A) shows the formation of a complex with stoichiometry 2:1 (Gos-iron). The stoichiometry of such Gos-iron complexes was determined by Job’s method that shows a breaking point at a molar ratio of 0.5.

Further evidence on the correlation between iron and Gos was obtained by IR spectroscopy (Figure 6B). For the free Gos and its iron complex, the broad band located in the range of 3000–4000 cm^−1^ was considered as the stretching of hydrogen-bonded hydroxyl groups due to the presence of water molecules. The ν (C = O) stretching mode of the free Gos occurs at 1654 cm^−1^, which has been shifted towards 1636 cm^−1^ upon complex formation. This result suggests that Fe(II) is correlated to carbonyl oxygen [31]. Moreover, the presence of the ν (Fe–O) stretching vibration at 630 cm^−1^ corroborates the formation of the iron-Gos complex, since free Gos does not exhibit such a band.

### 2.6. Gos Prevented Fe(III) Reduction by Ascorbate

The ferric state of iron promoted by Gos represents a plausible antioxidant mechanism, since it hinders the catalytic activity of Fe(II). Nevertheless, Fe(III) could still be reduced again to its ferrous form by natural reducing agents like ascorbate. The latter could reload biological systems with Fe(II), which participates in Fenton-Haber-Weiss reactions, generating the extremely reactive ^●^OH radical. To examine these possibilities, we used 1,10-phenanthroline to measure the levels of Fe(II) formation from a Gos solution (100 µM) treated with 2 mM ascorbate and different Fe(III) concentrations (10–100 µM). Figure 7 shows that the absence of Gos allowed a maximal reduction rate of Fe(III) to Fe(II) (black line with a slope of 5.85 × 10^−3^); however this process was slowed down by the presence of Gos (1-min incubation, green line), and after 5 min of incubation with Gos, the reduction rate of Fe(III) by ascorbate decreased almost 3 times (red line with a slope of 2.04 × 10^−3^). This result shows the capacity of Gos to inhibit the ascorbate-mediated Fe(III) reduction to Fe(II).

### 2.7. Gos Protects against 2-Deoxyribose Oxidative Degradation

To document the ability of Gos to act preferentially on iron instead of free radicals, competition studies were performed to evaluate the effectiveness of Gos and two ^●^OH scavengers (DMSO and salicylate) in protecting 2.8 or 28 mM 2-deoxyribose from iron-mediated oxidative damage (Figure 8). The ^●^OH scavengers at 20 mM protected 28 mM 2-deoxyribose significantly less than 2.8 mM 2-deoxyribose (*p* < 0.05), as expected. Gos was equally effective in preventing oxidative degradation of both 2.8 and 28 mM 2-deoxyribose.

## 3. Discussion

The iron released under several neurodegenerative conditions, including ischemic and hemorrhagic stroke, provokes the deregulation of brain iron homeostasis, leading to pathophysiology of neurological injury [4,32,33,34]. At subcellular levels, mitochondrial impairment seems to be involved in iron-mediated neuronal death [10,15,35,36,37,38]. Preclinical evidence supports the advantage of using iron chelators, mainly deferoxamine mesylate, against neurodegeneration including all types of stroke [7,16]. However, their high cost and unavailability, and the wide-range of adverse effects of classical iron chelators, have led to the use of natural chelators for iron management in dyshomeostasis [39,40].

The protective action of Gos is shown on HT-22 cells and rat-brain mitochondria incubated with an iron/citrate mixture, where we found a strong protection against iron-induced oxidative damage. Mouse hippocampal HT-22 cells exposed to iron overload (24 h) showed decreased viability, which was closely associated to early mitochondrial membrane potential dissipation and ATP reduction (Figure 2A–C, respectively). This suggests that mitochondrial impairment contributes to neuronal lethality. Indeed, under our conditions, at least a portion of Fe(II) uploaded to neurons is expected to reach the organelles, since iron-induced neuronal death was preceded by the loss of mitochondrial function. Hence, it has been shown that iron overload induces mouse hippocampal HT-22 cell death and mitochondrial fragmentation [41,42]. Likewise, sustained iron exposure increases mitochondrial ROS levels in dopaminergic neuroblastoma SHSY5Y cells [43]. Furthermore, an extensive iron influx into the hippocampal neurons as well as mitochondrial damage were verified after the exposure to 100 µM ferrous iron [36]. Here we observed that the Gos-induced preservation of viability of the hippocampal neurons affected by iron is closely related to the conservation of their mitochondrial membrane potential and ATP levels.

As in intact cells, the direct mitochondrial exposure to ferrous iron provoked the extensive swelling of the organelle, dissipation of membrane potential, and the loss of ATP (Figure 3A–C, respectively). In this experimental setting, Gos was able to protect the iron-overloaded mitochondria, which was expressed by the preservation of the above-mentioned parameters. In this sense, it has been observed that mitochondria loaded with a micromolar concentration of iron underwent mitochondrial permeability transition pore opening and ΔΨ dissipation in a manner that is sensitive to iron chelation but not dependent on catalase antioxidant action [44]. Interestingly, the Mito-Tempo antioxidant was reported to protect against iron overload damage in hippocampal neurons by scavenging the mitochondrial superoxide anion radical and preserving mitochondrial morphological integrity and membrane potential [36]. We recently described the antioxidant effects of this flavonoid and its ability to protect PC12 cells against chemical hypoxia-induced death [29], where it was concluded that the free radical scavenging and antioxidant ability of Gos is partly involved in the protection against iron-mediated HT-22 and mitochondrial damage. However, the improved efficacy of Gos against iron-mediated lipoperoxidation versus *tert*-butyl hydroperoxide-mediated lipoperoxidation (Figure 4A,B, respectively) strongly suggests that its iron-interacting capacity is the main mechanism against iron-induced damage.

To characterize the Gos-Fe interaction further, several cell-free and mitochondria-free experiments were performed. We observed that when polyphenol was co-incubated with Fe (II), the concentration of the metal ions decline, while there was a corresponding increase in oxygen consumption rate (Figure 5A–C). These effects suggest that Gos removes Fe(II) from the citrate complex, and oxidizes it to a ferric form in a process that requires O_2_ as an electron acceptor. Consequently, Gos promotes the decline of Fe(II) concentration that may hinder the hydroxyl radicals through Fenton reactions. Since the Gos-Fe(III) complex enables the oxidation of relevant reducing agents such as ascorbate, resulting in the formation/regeneration of Fe(II), we were able to demonstrate that Gos inhibits the ascorbate-mediated reduction of Fe(III) to Fe(II).

The hypothesis that Gos strongly interacts with iron was also here confirmed through spectroscopic techniques, and corroborated previous results where different arrays of Gos-iron complex stoichiometry were characterized by electro spin ionization mass spectroscopy [45].

These results evidenced the protective effects of Gos against iron-mediated neuronal damage, probably by interacting with ferrous ions, hindering its involvement in the catalytic reactive oxygen species formation. Furthermore, this suggests the formation of a transient charge-transfer complex between Fe(II) and Gos, accelerating Fe(II) oxidation and the formation of a more stable Fe(III)-Gos complex that cannot participate in the propagation phase of lipid peroxidation. Moreover, a biologically relevant reducing agent such as ascorbate was unable to reduce ferric iron in the presence of Gos, limiting a pro-oxidant characteristic of certain flavonoids involved in ferrous ion re-cycling. [30]. Gos at micromolar concentrations was more effective than classical ^●^OH scavengers in preventing iron-mediated oxidation of 2-deoxyribose. This high efficacy may be adscribed to the formation of a redox-active Gos–Fe(II)/(III) as we previously observed for others polyphenols that improved their performance as antioxidants upon their interaction with iron in different in vitro paradigms of oxidative damage [22,46,47,48].

It has been established that chelating agents that contain oxygen as a ligand (oxo ligand, O^2−^) can chelate iron and promote the oxidation of Fe(II), stabilization of Fe(III), and consequently produce a decrease in its reducing potential [49]. At physiological pH, catechols readily form thermodynamically stable bis complexes with ferric iron, favored by low concentrations of the ligands. The presence of a catechol moiety in the Gos structure suggests a similar interaction mechanism with iron, which could explain the protection achieved against iron-induced damage of neuronal cells and mitochondria. In this regard, we also previously demonstrated that mangiferin and guttiferone A stimulate ferrous iron oxidation and hinder ferric iron reduction [23,24,26,27,28].

The ability of Gos to chelate iron may also prompt signalling pathways that contribute to neuroprotection. For example, prolyl hydroxylase domain enzymes (PHD), the classical Hypoxia Inducible Factor-1alpha (HIF-1α) hydroxylation-modifying enzymes under normoxia, have been identified as the critical targets of iron chelators that are clinically beneficial in many neurological disorders [50,51]. It is understandable since PHD depend on divalent iron as a coupling factor [52]. Several genes that have been involved in neuroprotection are regulated by HIF-1a, such as eNOS, VEGF, and EPO [53]. Moreover, HIF activation resulting from PHD inhibition could prevent oxidative stress-mediated mitochondrial impairment and apoptosis, independently of its role as a transcription factor [54].

Ferroptosis, the recently characterized iron-dependent regulated cell death, has been proposed as the mechanism through which the neurons exposed to hemorrhagic damage die [8]. Mitochondrial dysfunction was recently related to ferroptotic cell death [55]. Therefore, the inhibition of ferroptosis, which saves mitochondrial function from iron damage, could also be a plausible mechanism for neuroprotection against iron-mediated neurological disorders protected by Gos, which would deserve further attention. Additional research on the presumed beneficial effects of Gos on in vivo animal models of iron-associated neurodegeneration must be made, in order to propose this flavonoid as a therapeutic intervention against brain tissue damage induced by iron homeostasis deregulation.

## 4. Materials and Methods

### 4.1. Reagents

All reagents were obtained from Sigma-Aldrich Corp. (St. Louis, MO, USA). Stock solutions of Gos were prepared in DMSO and added to the cell cultures or mitochondrial preparations at 1/1000 (*v*/*v*) dilutions.

### 4.2. Plant Material

Plant materials (flowers) from *Talipariti elatum* (Sw.) (Syn. *Hisbiscus elatus* Sw.) (*Malvaceae*) were collected during March 2020 from the surrounding areas of the Center for Research and Biological Evaluation, Institute of Pharmaceutical and Food Sciences, La Lisa, Havana, Cuba. They were authenticated by specialists of the herbaria of the University of Camagüey “Ignacio Agramonte Loynaz”, where voucher specimens were deposited: G. Pardo., HPC-12 520 (HIPC).

### 4.3. Extract and Sample Preparation

The ethanolic extracts were prepared from the red flowers of *T elatum* as previously reported [29]. After rotoevaporation, solids were purified by re-crystalization procedures [29]. The isolated compounds were characterized by means of 1D ^1^H (400 MHz) and ^13^C (100 MHz) NMR. NMR data processing was performed with the MestReNova software version 6.1.0-6224 for Windows (Appendix A). NMR data collected from Gos was compared with previous results [56].

### 4.4. UV/Vis and IR Spectroscopic Studies

UV/Vis spectra were performed in quartz cuvettes of 10 mm optical path length and absorbance was recorded in a UV-Vis spectrophotometer Ultrospec 2100 Pro from Amersham BioSci (Little Chalfont, UK) coupled to a computer with Wave Scan Software (SWIFT II, 1.2) at room temperature. The interaction between Gos and ferrous ions were measured by monitoring the changes in the UV/Vis spectrum. Job’s method was used to determine the stoichiometric ratio between Gos and the Fe(II) ion. The stock solutions of Gos and Fe(II) were prepared daily in methanol and phosphate buffer, respectively. The IR spectra were recorded on a JASCO FT/IR-440 spectrometer over the range of 4000–400 cm^−1^, using potassium bromide pellets to prepare the samples.

### 4.5. Cell Cultures

Mouse hippocampal HT-22 cells (ATCC) were cultured in a DMEM medium supplemented with 10% FCS, L-glutamine (1 mM), and penicillin/streptomycin (1%), and maintained at 37 °C in a humidified 5% CO_2_ incubator. The cells (2 × 10^4^ cell/well) were seeded into 96-well plates (Nunc, Roskilde, Denmark), in 200 μL of culture medium for 24 h at 37 °C.

### 4.6. Induction of Cell Oxidative Damage and Treatment with Gos

The neuroprotective action of Gos (0.001–100 μM) was tested on HT-22 by means of co-incubation for 24 h with Gos or Trolox (500 µM), FeCl_2_ (100 µM), and citrate (1 mM). After 24 h, the cell supernatant was removed and cell viability was measured by the [3-(4,5-dimethylthiazol-2-yl)-2,5-diphenyl tetrazolium bromide] (MTT) colorimetric assay.

### 4.7. Cell Viability Assay

After the treatments, cells (2 × 10^4^) were incubated with 10 μM MTT for 3 h at 37 °C. The coloured MTT derivative was then dissolved in 50 μL of DMSO, and the absorbance was counted at 570 nm on a microplate reader (FLUOstar Omega, BMG LABTECH, Ortenberg, Germany). Cell viability is expressed as a percentage against a control of untreated cells.

### 4.8. Mitochondrial Membrane Potential (ΔΨ) Assay in HT-22 Cells

Mitochondrial membrane potential was assessed on the basis of cell retention of the fluorescent cationic probe rhodamine 123 [57,58]. After 2 h of iron/ascorbate injury in the absence (control) or presence of Gos (0.001–100 μM) or Trolox (500 µM), the neuronal cells (2 × 10^4^) were incubated 10 min with 1 μM rhodamine 123. After centrifugation, and resuspension in 1 mL of 0.1% Triton X-100, the probe concentration was determined in the supernatant using a microplate reader (FLUOstar Omega, BMG LABTECH, Ortenberg, Germany) at the 505/535 nm excitation/emission wavelength pair. Data are expressed as percentages versus the Control of untreated cells.

### 4.9. ATP Assay

Cellular ATP was determined by the firefly luciferin/luciferase assay system [59] under the same conditions as ΔΨ. Bioluminescence was measured with a Sigma-Aldrich assay kit according to the manufacturer’s instructions, using the luminescence option of the microplate reader (FLUOstar Omega, BMG LABTECH, Ortenberg, Germany).

### 4.10. Animals

Male Wistar rats (200 g) were obtained from the Center for the Production of Laboratory Animals (CENPALAB according to its Spanish acronym, Havana, Cuba) and were housed in the animal care facility for 1 week prior to experiments in a temperature-controlled environment (22–24 °C), with a 12-h light/dark cycle and with access to food and water ad libitum.

### 4.11. Mitochondrial Isolation

Rat forebrain mitochondria were isolated by differential centrifugation as described previously [60]. Digitonin 10%, was used to disrupt synaptosomes and yield a mixture of non-synaptosomal plus synaptosomal mitochondria. The entire procedure was carried out at 4 °C. The respiratory control ratio (state 3/state 4 respiratory rate) was always in the 4.5 to 6.0 range, measured using 5 mM succinate as the substrate.

### 4.12. Continuous-Monitoring Mitochondrial Assays

The mitochondrial membrane potential was determined spectrofluorimetrically using 10 µM safranine O as a probe [58,61] in a microplate reader (FLUOstar Omega, BMG LABTECH, Ortenberg, Germany) at 495/586 nm excitation/emission wavelengths; these assays were performed in the presence of 0.1 mM EGTA and 2 mM K_2_HPO_4_. Mitochondrial swelling was estimated spectrophotometrically from the decrease in apparent absorbance at 540 nm. For the assays, mitochondria were energized with 5 mM potassium succinate (plus 2.5 µM rotenone) in a standard medium consisting of 125 mM sucrose, 65 mM KCl, and 10 mM HEPES-KOH, pH 7.4, at 30 °C.

### 4.13. Lipid Peroxidation Assays

Lipid peroxidation was estimated from malondialdehyde (MDA) generation [28,62]. Mitochondria (1 mg/mL) were exposed to 4 mM tert-butylhydroperoxide or 50 µM Fe(II) plus 1 mM sodium citrate at 37 °C. MDA concentration was calculated by using its extinction coefficient (ε = 1.56 × 10^5^ M^−1^·cm^−1^).

### 4.14. Determination of Fe(II) Concentration

The concentration of Fe(II) was determined in 10 mM phosphate buffer pH 6.5 (2 mL) by the formation of a red complex with 1,10-phenanthroline (5 mM), as previously described [24,28].

### 4.15. Oxygen Consumption Monitoring

The consumption of O_2_ related to Fe(II) oxidation was measured with a Clark-type electrode (Hansatech Instruments, Pentney, King’s Lynn, UK) in a 1.3-mL glass chamber equipped with a magnetic stirrer, at 30 °C. The rate of oxygen consumption (ROC) was expressed as nmol O_2_/mL per min.

### 4.16. 2-Deoxyribose Oxidation Assay

The formation of hydroxyl (^●^OH) radicals was estimated by following the 2-deoxyribose degradation to MDA, which was calculated from its extinction coefficient (ε = 1.56 × 10^5^ M^−1^·cm^−1^) as previously reported [26,27].

### 4.17. Gos-Fe(II) Complexes Detection

Fe(II) (FeCl_2_, 0.4–1.6 μM) was added to 2 mL (10 mM) phosphate buffer (pH 6.5) containing 2 mM citrate and 1.6 μM Gos. A wavelength scan from 200–600 nm was performed in a Hitachi 2001 spectrophotometer (Hitachi, Tokyo), at 28 °C.

### 4.18. Statistical Analysis

GraphPad Prism 5.0 (GraphPad Software, Inc., San Diego, CA, USA) was used. Statistical analysis was performed using a one-way ANOVA, followed by a Student-Newman-Keuls post-hoc test for pair-wise comparisons. Results with *p* < 0.05 were considered statistically significant. The half-effective or inhibitory concentrations were estimated following a non-linear regression algorithm.

## Figures and Tables

**Figure 1 molecules-26-03364-f001:**
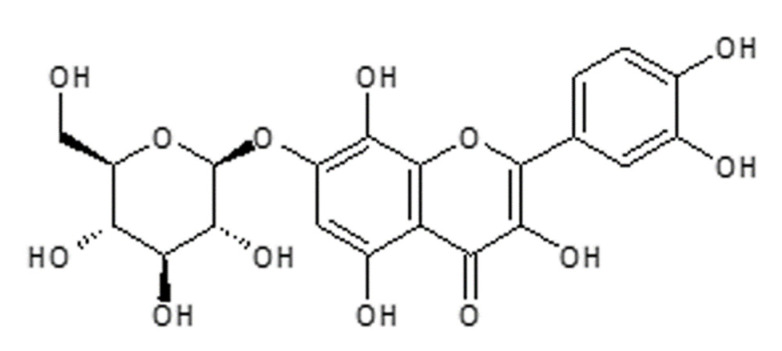
Gossypitrin (Gos) structure. 2-(3,4-Dihydroxyphenyl)-3,5,8-trihydroxy-4-oxo-4H-chromen-7-yl β-d-glucopyranoside.

**Figure 2 molecules-26-03364-f002:**
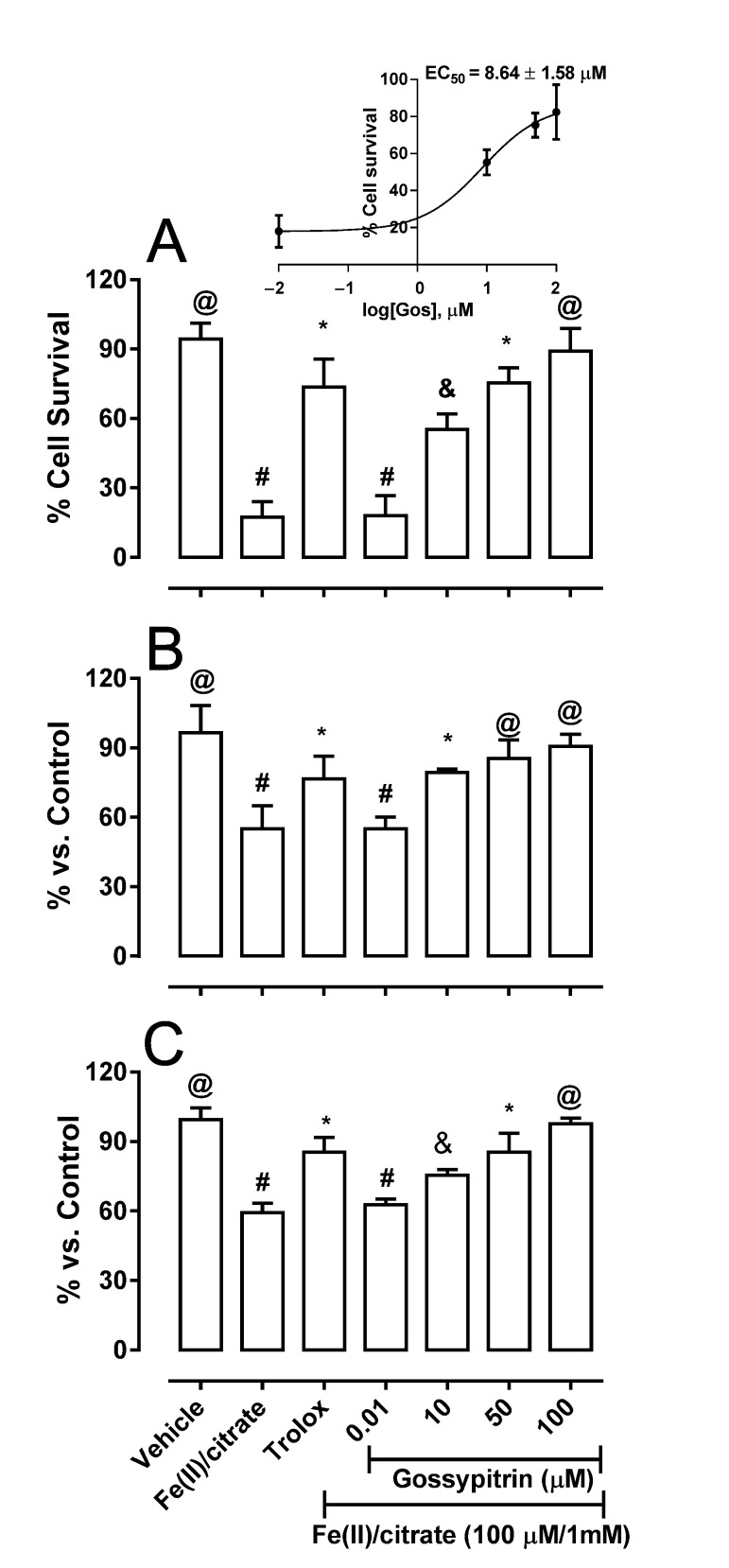
Gos protection against iron-induced neuronal cell damage. Effects of Gos on HT-22 mouse hippocampal neuron survival (**A**), cellular mitochondrial membrane potential (**B**), and cellular ATP levels (**C**). Gos (0–100 μM) or Trolox (500 µM) were co-incubated for 24 h (viability) or 2h (mitochondrial membrane potential and ATP levels) with 100 µM FeCl_2_/1 mM citrate. The assay conditions are described in Section 4. Damaged control cells (without Gos) contained DMSO (0.001%) plus the iron-citrate mixture. The results are expressed as the percentage of cell death in relation to the undamaged control (containing DMSO 0.001% as the vehicle). Values are the means ± S.D. of three different experiments. Different symbols indicate statistical differences at *p* < 0.05 when comparing all pairs of experimental groups according to the ANOVA and post hoc Newman-Keuls tests. (If the symbols are different among groups indicate statistical differences, when they are equal, there is not).

**Figure 3 molecules-26-03364-f003:**
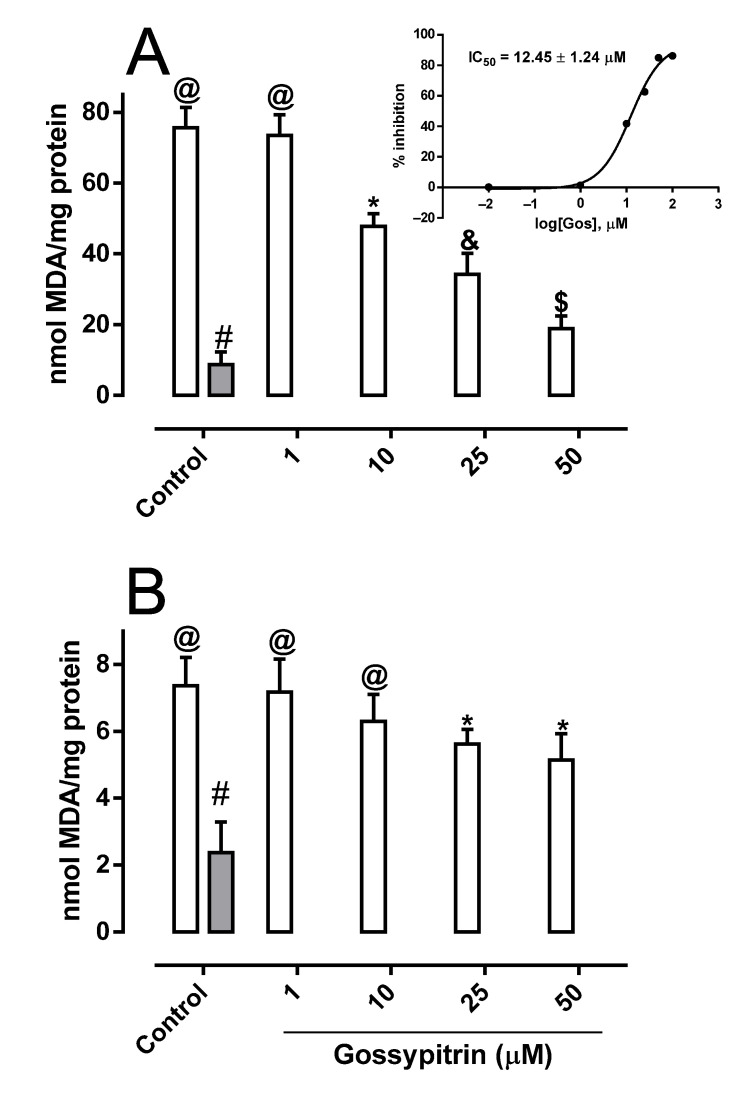
Gos inhibits malondialdehyde formation in the mitochondria, induced by 50 µM Fe(II)-citrate (**A**) or 300 µM *tert*-butyl hydroperoxide (**B**). Rat brain mitochondria (1 mg/mL) were incubated in a reaction medium containing 125 mM sucrose, 65 mM KCl, 10 mM HEPES buffer (pH 7.4), 2 mM succinate, and 2.5 µM rotenone, with or without Gos (1–50 µM). The experiments started by the addition of 50 µM Fe(II) or 300 µM *tert*-butyl hydroperoxide, except for the undamaged Control (gray bars). The incubation period was 20 min at 28 °C. Values are the means ± S.D. (*n* = 6). Different symbols indicate statistical differences at *p* < 0.05 when comparing all experimental groups according to the ANOVA and post hoc Newman-Keuls tests. (If the symbols are different among groups indicate statistical differences, when they are equal, there is not).

**Figure 4 molecules-26-03364-f004:**
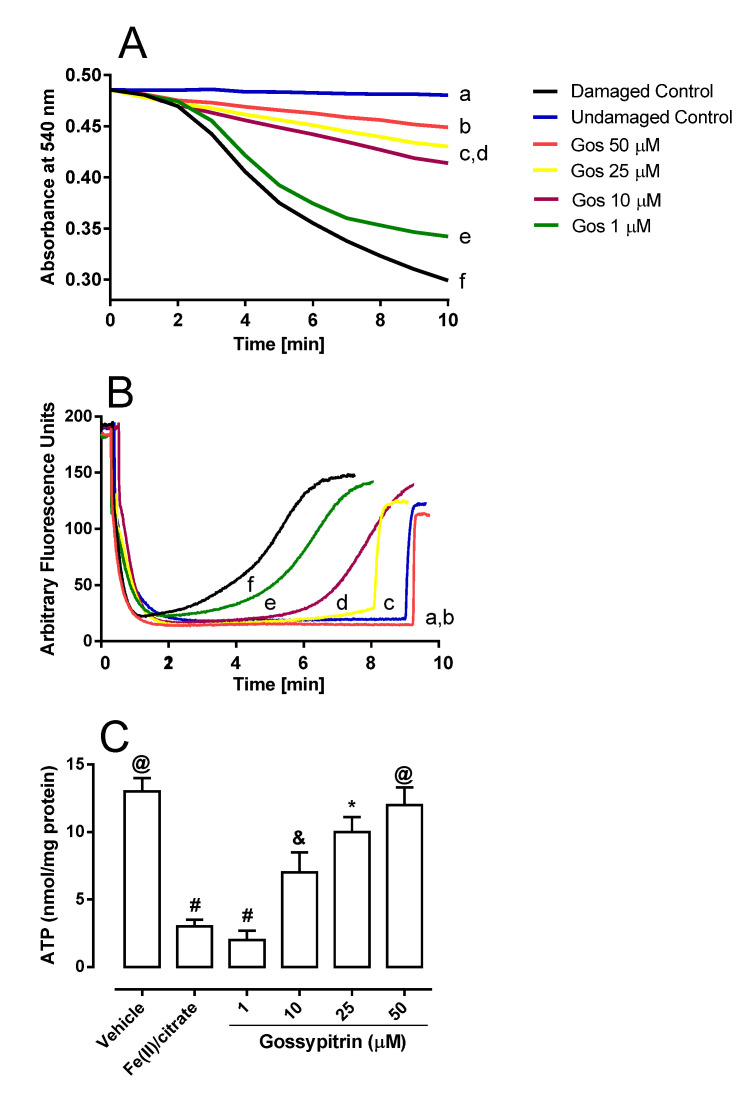
(**A**) Mitochondrial permeability transition induction, (**B**) mitochondrial membrane potential (ΔΨ) loss, and (**C**) reduction of ATP levels induced by 50 µM Fe(II)-citrate. ΔΨ was estimated using the fluorescence probe safranine (5 µM). The inhibitory effects of Gos 50, 25, 10, and 1 µM are shown in lines (b), (c), (d), and (e), respectively. Lines (a) and (f) represent undamaged controls (without iron), and damaged controls (without Gos), respectively. Rat brain mitochondria (RBM-0.5 mg/mL), Fe(II) and 1 µM CCCP were added where indicated by the arrows. Different symbols in graph (**C**) indicate statistical differences at *p* < 0.05 when comparing all pairs of experimental groups according to the ANOVA and post hoc Newman-Keuls tests. (If the symbols are different among groups indicate statistical differences, when they are equal, there is not).

**Figure 5 molecules-26-03364-f005:**
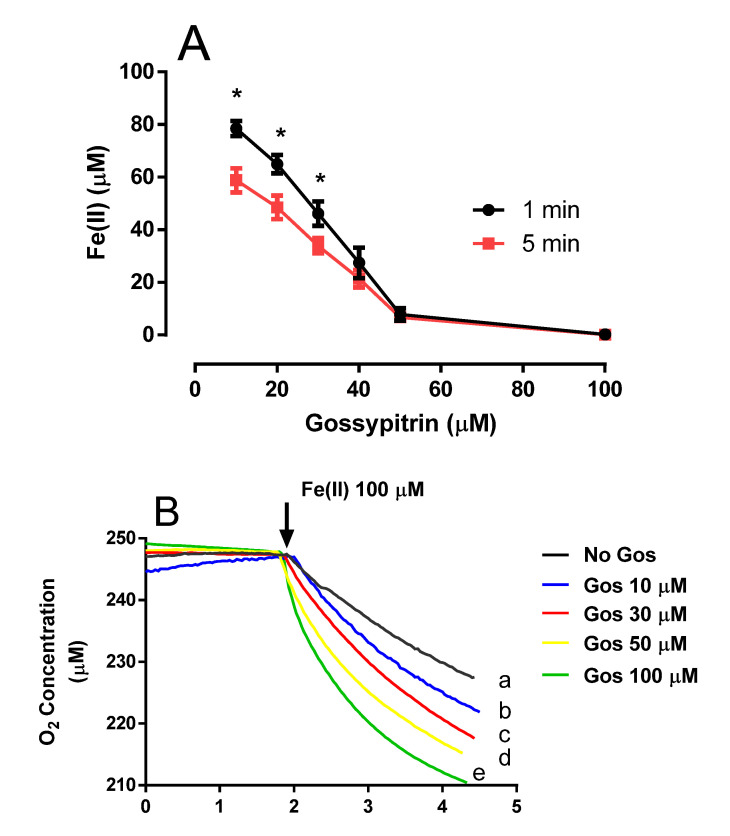
(**A**) Gos stimulates Fe(II) autoxidation in the absence of rat brain mitochondria in a standard medium supplemented with 1 mM citrate, and 5 mM 1,10-phenanthroline. 1,10-phenanthroline was added 1 or 5 min after the Gos–Fe(II) incubation period, and the absorbances were read at 510 nm. Values are the average of three determinations. * statistical difference at *p* < 0.05 when comparing 1 vs. 5 min incubation. (**B**) Effects of Gos on O_2_ consumption mediated Fe(II) autoxidation in a standard medium supplemented with 1 mM citrate under the following conditions: (**a**) No Gos, (b) 10 µM Gos, (c) 30 µM Gos, (d) 50 µM Gos, and (e) 100 µM Gos. Fe(II) (50 µM) was added where indicated by the arrow. Results are representative of three experiments. (**C**) Oxygen consumption rate (nmol O_2_/min/mL) 1 min after Fe(II) addition. Legends are the same as in (**B**). Different symbols indicate statistical differences at *p* < 0.05 when comparing all pairs of experimental groups (in the same cell type) according to the ANOVA and post hoc Newman-Keuls tests. (If the symbols are different among groups indicate statistical differences, when they are equal, there is not).

**Figure 6 molecules-26-03364-f006:**
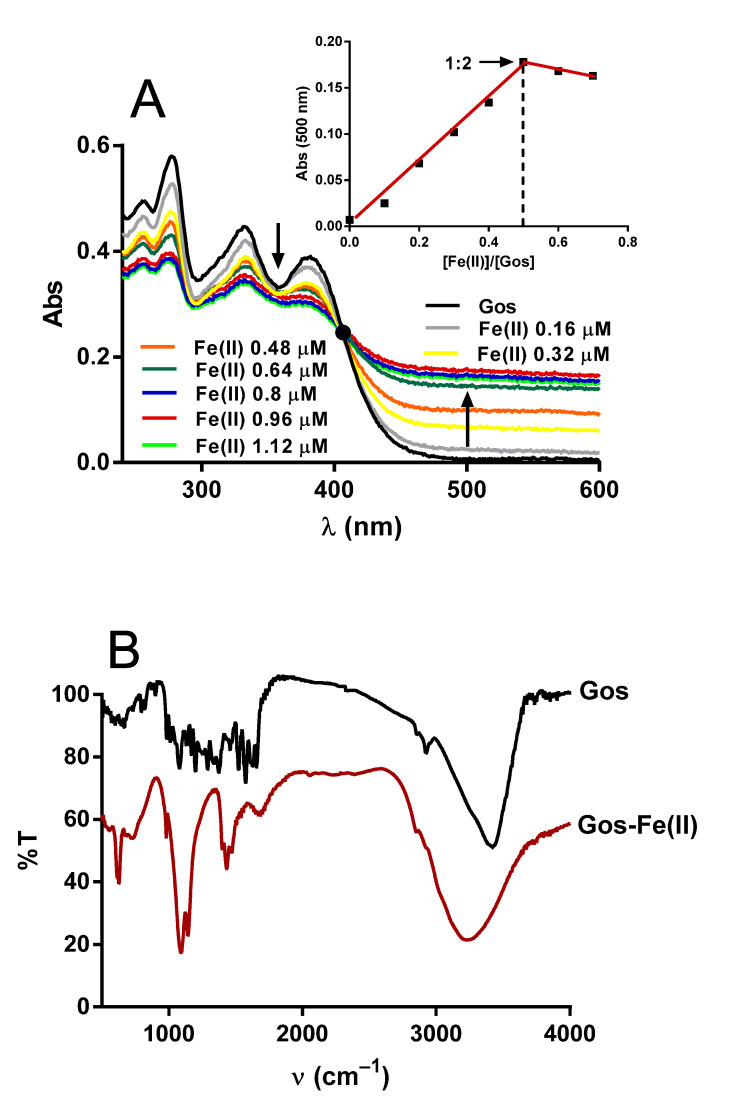
(**A**) Effects of Fe(II) on the Gos UV–VIS spectrum (200–600 nm). Incubation mixture containing 125 mM sucrose, 65 mM KCl, 10 mM HEPES buffer (pH 7.2), 2 mM citrate, and Gos 1.6 µM. The concentrations of Fe(II) from top to bottom traces were 0, 0.16, 0.32, 0.48, 0.64, 0.8, 0.96, and 1.12 µM. Inset: Job’s plot for the Gos-Fe(II) complex at a constant total concentration [Fe(II)] + [Gos] = 1.6 μM. Black dot indicates the isosbestic point. The downward and upward arrows indicate decrease and increase in absorbance values at 380 and 500 nm, respectively. Experiments were conducted at 28 °C. Scan speed was 2 nm/s. A baseline was established with the incubation mixture plus 1.6 µM Fe(II). (**B**) Infrared spectra of Gos and Gos-Fe(II). Typical examples are shown.

**Figure 7 molecules-26-03364-f007:**
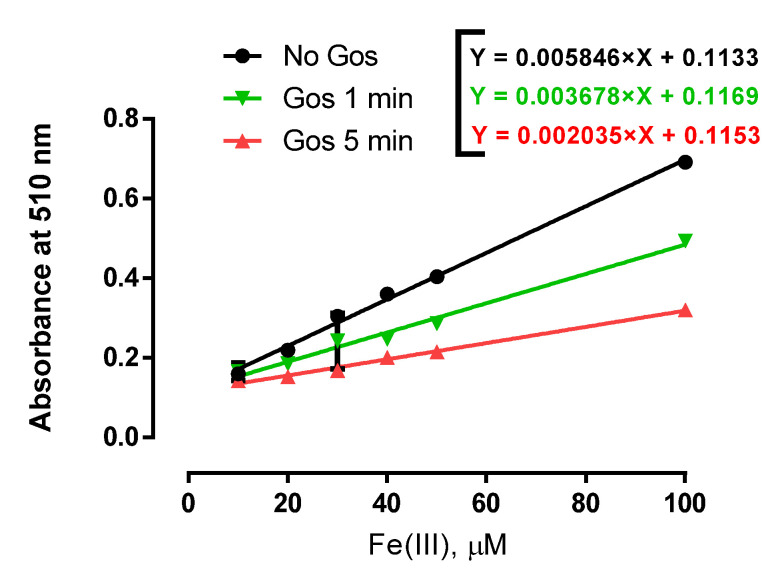
Gos inhibits Fe(III)-reduction by ascorbate in the absence of rat liver mitochondria. Experimental conditions: 125 mM sucrose, 65 mM KCl, 10 mM HEPES buffer (pH 7.2), 1 mM citrate, Gos 100 µM. Experiments were conducted at 28 °C. Ascorbate (4 mM) and 5 mM 1,10-phenanthroline were added after 1 min or 5 min of Gos-Fe(III) incubation. Lines are representative of three assays.

**Figure 8 molecules-26-03364-f008:**
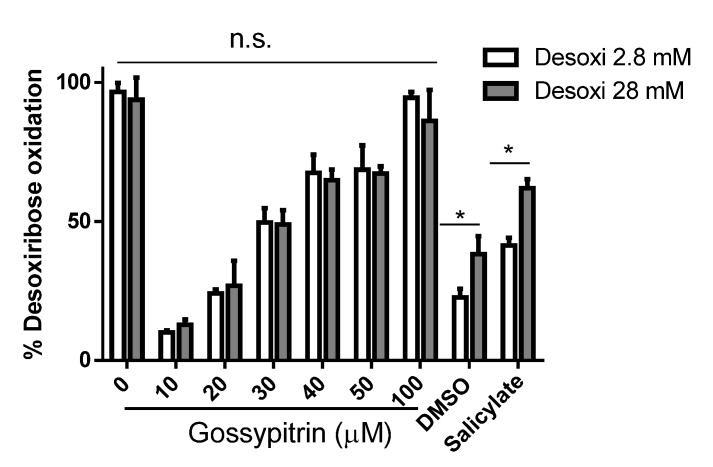
Effect of Gos and •OH scavengers dimethyl sulfoxide (DMSO) and salicylate on oxidative damage to 2.8 or 28 mM 2-deoxyribose induced by Fe(III)–EDTA plus ascorbate. Solutions were incubated for 30 min at 37 °C and contained 10 mM phosphate buffer (pH 7.2), 2-deoxyribose (2.8 or 28 mM), 150 µM EDTA, and 50 µM Fe(III). Reactions were started by the addition of ascorbate to a final concentration of 2 mM. The bars show means ± S.D. (*n* = 3). Controls contain only DMSO (0.001%), which is the solvent concentration in Gos samples. The one-tailed *t*-test was used for statistical analyses; * *p* < 0.05, n.s., non-significant.

## Data Availability

Not applicable.

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
