# Peer review of "Gossypitrin, A Naturally Occurring Flavonoid, Attenuates Iron-Induced Neuronal and Mitochondrial Damage"

_molecules, 2021, doi:10.3390/molecules26113364_

Round 1
Reviewer 1 Report
Thanks for the opportunity to review this manuscript. The subject is really important because of the growing number of patients with neurological diseases, which carries a large socio-economic problems. A study identifying a new, natural substances to support the treatment of these diseases which are extremely relevant. Nevertheless, in my opinion, a few issues should be clarified before the publication of the article.
- I have the biggest reservations about the discussion. In my opinion, this is a more detailed description of the obtained results than a methodical discussion of the obtained results with other studies. Moreover, for me, the work would be more understandable if it included a description of the signaling pathways explaining the mechanism of action of gossypitrin.
- The language is incomprehensible in some places, linguistic correction is required
- The quality of the figures and signatures is not very satisfactory
Reviewer 2 Report
The manuscript entitled “Gossypitrin, a naturally occurring flavonoid, attenuates iron-induced neuronal and mitochondrial damage” by Bécquer-Viart et al. reports the Gossypitrin ability to protect from neuronal diseases associated with iron-induced oxidative stress and mitochondrial damage.
In the introduction the topics is properly illustrated. The authors reported a detailed biological section supported by a good rational and all experiments are properly described.
The manuscript is well written, and suitable for publication in Molecules after revision of few points:
Introduction:
“The presence of a planar six-member cyclic system with electron delocalization, a catechol moiety, and three more aromatic hydroxyl groups (two of them adjacent to a carbonyl group), strongly suggest that this molecule could act as an iron-chelating agent”
My suggestion is to focus on this point supporting it with the literature. Are there any references in literature about the iron-chelating properties of structures that are similar to Gos?
Line 49: There is a missing bracket
Results:
The presence of letters on the histograms is confusing. My suggestion is to insert a legenda in the caption of each figures and/or to remove the letters where there are not necessary.
Figure 6B: the comparison of IR spectra of Gos and Gos-Fe is described. It could be interesting to compare also the 1HNMR of these two samples.
Round 2
Reviewer 1 Report
Thank you for responding to my comments and for revising the manuscript.